# Simultaneous Quantification of Bisphenol-A and 4-Tert-Octylphenol in the Live Aquaculture Feed *Artemia franciscana* and in Its Culture Medium Using HPLC-DAD

**DOI:** 10.3390/mps5030038

**Published:** 2022-05-01

**Authors:** Despoina Giamaki, Konstantina Dindini, Victoria F. Samanidou, Maria Touraki

**Affiliations:** 1Laboratory of General Biology, Division of Genetics, Development and Molecular Biology, Department of Biology, School of Sciences, Aristotle University of Thessaloniki (A.U.TH.), 54 124 Thessaloniki, Greece; despina_giamaki@imbb.forth.gr (D.G.); konstantina.dindini@stud.ki.se (K.D.); 2Laboratory of Analytical Chemistry, Department of Chemistry, School of Sciences, Aristotle University of Thessaloniki (A.U.TH.), 54 124 Thessaloniki, Greece; samanidu@chem.auth.gr

**Keywords:** bisphenol A, 4-tert-octylphenol, *Artemia franciscana*, HPLC-DAD

## Abstract

Aquaculture, a mass supplier of seafood, relies on plastic materials that may contain the endocrine disruptors bisphenol-A (BPA) and tert-octylphenol (t-OCT). These pollutants present toxicity to *Artemia*, the live aquaculture feed, and are transferred through it to the larval stages of the cultured organisms. The purpose of this work is the development and validation of an analytical method to determine BPA and t-OCT in *Artemia* and their culture medium, using n-octylphenol as the internal standard. Extraction of the samples was performed with H_2_O/TFA (0.08%)–methanol (3:1), followed by SPE. Analysis was performed in a Nucleosil column with mobile phases A (95:5, *v*/*v*, 0.1% TFA in H_2_O:CH_3_CN) and B (5:95, *v*/*v*, 0.08% TFA in H_2_O:CH_3_CN). Calibration curves were constructed in the range of concentrations expected following a 24 h administration of BPA (10 μg/mL) or t-OCT (0.5 μg/mL), below their respective LC_50_. At the end of exposure to the pollutants, their total levels appeared reduced by about 32% for BPA and 35% for t-OCT, and this reduction could not be accounted for by photodegradation (9–19%). The developed method was validated in terms of linearity, accuracy, and precision, demonstrating the uptake of BPA and t-OCT in *Artemia*.

## 1. Introduction

The worldwide accumulation of plastic pollution in the marine environment poses a great risk towards the wellbeing of the vertebrate and invertebrate organisms that live in it, as well as of humans as the end consumer. The protective gear that was extensively produced during the COVID-19 pandemic, has greatly contributed to the plastic waste, a large amount of which ends up in the marine environment, where leaching occurs of its hydrophobic additives [1]. The highly hydrophobic organic contaminants include endocrine disrupting chemicals (EDCs), such as bisphenol A (BPA), nonylphenol, and tert-octylphenol (t-OCT). EDCs are incorporated into plastics as building blocks or stabilizers [2] and leach out, due to aging and heat [3,4]. Bisphenol-A (4, 4′-Isopropylidenediphenol) is widely used as a component of synthetic plastic [2] while 4-tert-octylphenol (4-(1,1,3,3-tetramethylbutyl) phenol) is used as a component of polyethoxylates, applied in detergents, industrial cleaners, and emulsifiers [3]. The percentage of stabilizers and antioxidants, including BPA and t-OCT in plastics, depends on the chemical structure of the produced plastic polymer, ranging from 0.05 to 3% *w/w* [5]. Studies on the levels of BPA and t-OCT in microplastics are gaining attention, and marine microplastics sampled in the open Pacific and the Atlantic Ocean, as well as from beaches in Asia and Central America, reported concentrations reaching 1–730 ng/g for BPA, and 0.1–153 ng/g for t-OCT [6]. BPA has been listed as one of the very high concern compounds by the European Chemicals Agency [7]. The European Food Safety Authority (EFSA) reassessed in 2015 the current tolerable daily intake (TDI) for BPA and reduced it from 50 to a temporary value of 4 μg /kg b. w. daily [8]. However, the plastic industries are in a legal dispute against the European Chemicals Agency’s (ECHA) decision to identify bisphenol A (BPA) as a “substance of very high concern” [9]. It appears that, although BPA is banned in some countries such as Canada, the fight for BPA to stay on shelves continues in Europe [10]. On the other hand, t-OCT is under assessment as persistent, bio—accumulative, and toxic, especially to aquatic life [11,12]. Both compounds present endocrine disrupting properties, resulting from their structural resemblance to the human 17β-estradiol [13], with t-OCT presenting higher estrogenicity than the other pollutants [14,15].

Following uptake, BPA is metabolized through its transformation to glucuronide and sulfate derivatives by vertebrates, such as tadpoles and fish [16], as well as mammals [17], while bacteria lead to the biotransformation of BPA to Hydroquinone, 4-Hydroxyacetophenone, 4-Hydroxybenzoic acid, and 4-Isopropenylphenol [18]. Although BPA has been detected in invertebrates, the reports on its metabolism by aquatic invertebrates are available only for bivalves, which transform it to mono- and disulfate [19]. In a similar manner, alkylphenols, including t-OCT, are metabolized by glucuronidation and/or sulfation in the liver of mammals such as rats [20] and humans [21], while bacteria transform it to end products including hydroquinone, 4-Hydroxybenzaldehyde and 4-Hydroxybenzoic acid [22].

Endocrine disruptors, such as BPA, induce oxidative stress and exert toxic effects on freshwater aquatic organisms by altering bacteria composition in their environment [23]. In the marine environment, sorption of EDCs from plastic particles in the surrounding seawater has been documented, indicating that marine plastics, under environmentally induced stress conditions, can act as carriers of organic contaminants and transfer them to marine organisms [3]. Marine organisms ingest the microplastics together with their contaminants. Amounts of BPA have been detected in fish [24], bivalves [25], and seawater [26], while t-OCT was detected in fish [27], mollusks [28], and shrimp [29]. Nowadays, the largest portion of all seafood that are used as food for humans, originate from aquaculture [30]. Aquaculture relies on plastic use in many aspects, such as fish cages, fish feeders, and fish tanks [31,32]. Moreover, aquaculture hatcheries require the live feed *Artemia* nauplii or metanauplii for feeding of the cultured fish and crustacea larvae [33,34], due to its high nutritional value characterized by high contents of neutral lipids [35]. Although *Artemia* is a crustacean adaptable to a wide range of environmental conditions, it is absent in common marine ecosystems, with its natural habitat being high salinity environments such as salt lakes, where it is found in the form of dormant cysts [36]. This saltwater planktonic crustacean, being a filter-feeder, consumes small-sized particles such as microalgae or organic manure [37], and even microplastics that are present in the water column [38,39]. The transfer of pollutants from microplastics to *Artemia* nauplii, and then to the zebrafish that consumed these nauplii as a feed, has been demonstrated for benzopyrene [40]. The genus *Artemia* spp. is widely used as a toxicity testing model due to its handling advantages, which include the fast hatching of cysts to nauplii, metanauplii, or adults, and hence the ease of access to these developmental stages [41,42]. Moreover, BPA toxicity against the first developmental stages (Instar I-II) of *Artemia franciscana* nauplii [43,44] as well as of the alkylphenol n-hexylphenol against *A. sinica* [45] have been reported, without, however, including t-octylphenol or a determination of the compounds in *Artemia*. The transfer of the endocrine disruption across species from the freshwater planktonic crustacean *D. magna* to both its consumer organisms, namely fish [46], as well as to its prey that is the algae it feeds on [47], was considered an indication of a broader action of endocrine disruptors that extends beyond the target organisms, further emphasizing the ecological risk.

Bisphenol-A has gained the focus of attention and the determination of BPA in food [48], and environmental samples [49] has been extensively reviewed. The reported methods mostly employ solvent extraction and SPE (polymer, Oasis, or C18) for the isolation of BPA from samples, followed by HPLC analysis (C18 columns). Moreover, the determination of BPA has been reported in biological samples such as rat tissues by HPLC, employing different extraction protocols for the serum and tissues and using a C18 column and an acetonitrile and water mobile phase with gradient elution [50]. In human breast milk, BPA was extracted using Matrix Solid Phase Dispersion (LiChrolut cartridges) and reversed phase HPLC, followed with isocratic elution of a C18 column with acetonitrile-water (70:30, *v*/*v*) [51]. The determination of the residual monomers, including BPA, released from resin-based dental restorative materials employed HPLC analysis in a Kromasil 100-C18 column eluted with methanol: acetonitrile: water, 60:15:25%, *v*/*v* [52], while the same HPLC analysis in saliva samples was performed using a Perfect Sil Target ODS-3 column eluted with acetonitrile/water, 58/42% *v*/*v* [53,54]. Fewer studies are engaged in the determination of multiple endocrine disruptors in complex biological samples, since extensive clean-up due to matrix interferences is required and only trace levels of the compounds are present. To this end, elaborate techniques including microwave-assisted extraction, C18 SPE, derivatization, and GC analysis were employed for the simultaneous determination of steroid EDCs, t-OCT, 4-cumylphenol, 4-nonylphenol, and BPA in fish samples [55]. The determination of BPA either alone or in combination with 4-t-octylphenol and 4-nonylphenol in human media has been thoroughly reviewed, and was performed in blood serum by ELISA or RIA, in plasma and urine by LC equipped with a fluorescence or electrochemical detector and a C18 column, while more complex samples, such as semen and placental tissue, required the use of LC-MS using a Shodex column, or LC -MS/MS using a C18 column [56]. A recent report on the determination of BPA and other xenoestrogens including t-OCT in human urine, serum, and breast milk samples, employed liquid chromatography-tandem mass spectrometry (LC-MS/MS) method on a silica Acquity column [57]. Analysis of several emerging contaminants, including BPA but not t-OCT, was performed in poultry manure using ultrasound-assisted matrix solid-phase dispersion for the extraction of the analytes and analysis following derivatization was performed by GC coupled to tandem mass spectrometry [58]. Simultaneous determination of BPA and alkylphenols has recently been reported for fish and gull [27], water samples [59], mussels [60], and naturally occurring marine zooplankton [61], using ultrasonic extraction followed by SPE (Oasis HLB) and analysis was performed by HPLC -FLD using a HYPERSIL GOLD C18 column and gradient elution with acetonitrile: water. The methods for the simultaneous determination of BPA and certain alkylphenols mentioned above, require specific equipment including ultrasonic extraction, fluorescence, MS or MS/MS detectors, which are not always available. A simple HPLC method, for the determination of BPA and its metabolites in bacterial cultures, was previously developed [18,62] and employed filtering of the medium samples and SPE (C18) extraction of BPA form bacteria, while HPLC analysis was conducted on a C18 Nucleosil column with gradient elution (acetonitrile: water containing TFA). However, this method did not include t-OCT and was validated only regarding the determination of BPA and its metabolites in cultures of bacteria in minimal salt media with BPA as the main carbon source. The live fish feed *Artemia* nauplii and metanauplii are unique regarding their biochemical composition, habitat, and extensive use as a live larval feed in aquaculture. Artemia is not included in the naturally occurring zooplanktonic communities, since it lives in high salinity habitats [36]. Its developmental stages, the nauplii and metanauplii, contain high levels of protein and lipids and especially unsaturated fatty acids and their composition can be enhanced through their enrichment with the appropriate nutrients, a fact that renders them suitable to serve as a live feed of larval stages in aquaculture [63]. However, this particular composition of Artemia as well as of its culture medium, namely seawater 35 ppt, poses significant analysis problems due to matrix interferences. Although the transfer of organophosphorus pesticides [64] and nonylphenol [65] from the crustacean *Artemia* to fish has been documented, the risk of the transfer of BPA and t-OCT to the cultured larvae of marine organisms through their live feed has not been evaluated, possibly due to the scarcity of methods on the determination of bisphenol and t-octylphenol in *Artemia*.

In the present study, *Artemia franciscana* metanauplii were used to develop and validate an analytical method that would enable the quantification of BPA and t-octylphenol, in the organisms as well as in their culture medium. The application of this method demonstrates that both bisphenol and t-octylphenol are ingested by *Artemia* and allows their quantification.

## 2. Materials and Methods

### 2.1. Chemicals and Reagents

The organic solvents, methanol and absolute ethanol, were supplied by Fisher Scientific (Loughborough, UK), while acetonitrile was supplied by VWR Chemicals (Paris, France). Trifluoroacetic acid (TFA) and sodium hypochlorite solution 10% *w*/*v* were purchased from AppliChem GmbH (Darmstadt, Germany). The sea salt used as the *Artemia* culture medium was purchased from Instant Ocean (Blacksburg, VA, USA). *Artemia*
*franciscana* cysts were kindly supplied by INVE Aquaculture (INVE HELLAS SA, 93, Kyprou str., 16451 Argyroupoli, Athens, Greece). The C_18_ sorbent material, columns, and frits for SPE were supplied by Grace Davison Discovery Sciences (Bannockburn, IL, USA).

The internal standard 4-n-Octylphenol and Bisphenol-A were supplied by Alfa Aesar (Karlsruhe, Germany), while 4-tert-Octylphenol was obtained from Sigma-Aldrich (Germany). The investigated compounds are presented in Table 1.

### 2.2. Animals

Experiments were performed on *A. franciscana* (Kellogg) cysts (e.g., grade, GSL strain) provided by INVE (INVE HELLAS S.A., Athens, Greece). The cysts were stored at 4 °C until use. The axenic culture of *Artemia* cysts was performed following their hydration and decapsulation in hypochlorite solution, as previously described [72,73]. In all procedures the artificial sweater was autoclaved prior to use and the containers were rinsed with ethanol to ensure bacteria-free cultures. The nauplii (instar I) were cultured for a total of 96h at a density of 15 nauplii per mL and they were fed on processed yeast provided by P. Sorgeloos (Laboratory of Aquaculture and *Artemia* Reference Center, University of Ghent, Belgium). The sterile culture medium was renewed daily, and feeding was stopped prior to the addition of pollutants at 72 h. In the preliminary experiments, the LC_50_ was determined for each endocrine disruptor. To this end, two experimental series of nauplii were employed at 72 h since the onset of cyst incubation after being fasted for six hours, and the first series received increasing concentrations of bisphenol (0, 5, 10, 15, 20, 25 or 30 μg/mL) while the second series received increasing concentration of 4-tert-octylphenol (0, 0.25, 0.5, 0.75, 2, 4 or 6 μg/mL). Each series employed samples containing 0 μg/mL of endocrine disruptor to serve as controls. Survival and mortality were recorded following a 24 h exposure to the pollutant and LC_50_ was estimated using linear regression and Probit analysis.

### 2.3. Preparation of Stock and Standard Solutions

The stock solution of BPA (500 μg/mL) and of t-OCT (200 μg/mL) and of the internal standard n -octylphenol (500 μg/mL) were prepared in methanol. All stock solutions were kept in 4 °C, in glass volumetric flasks protected from light to avoid photodegradation of the analytes.

A set of six working standard solutions were prepared for BPA in two sets containing 2.5, 5, 7.5, 10, 12.5 and 15 μg/ mL or 1, 25, 50, 75, 100 and 125 μg/ mL. For t-OCT a set of six working standard solutions were prepared containing 1, 2.5, 5, 7.5 10 and 12.5 μg/mL. Each spiked sample contained the internal standard n-octylphenol at a concentration of 75 μg/mL. The standard solutions were transferred in amber glass vials (SU860083, Supelco) and stored at 4 °C. Calibration curves were prepared at six points, with concentrations ranging from 2.5 to 12.5 μg/mL for BPA in tissue, from 1 to 125 μg/mL for BPA in medium and from 1 to 12.5 μg/mL for t-OCT.

### 2.4. Sample Preparation

The protocol previously developed for the extraction of Bisphenol-A in bacterial cultures [61] was modified appropriately and validated to facilitate purification of both BPA and t-OCT in *Artemia* nauplii, as well as in their culture medium.

Regrading extraction of BPA and t-OCT from biological samples, 0.2 g wet weight of *A. franciscana* nauplii were homogenized (4 °C) in 2 mL methanol and 10.5 mL of double distilled water containing 0.08% (*v*/*v*) TFA, following the addition of the internal standard n-octylphenol (final concentration of 75 μg/mL). A Kinematica Polytron PCU homogenizer was employed for 5 min and the sample was kept in ice. The homogenate was centrifuged for 10 min at 10,000 rpm. The supernatant was submitted to SPE (0.5 g of C_18_ sorbent) conditioned with 10 mL methanol, 5 mL ddH_2_O, and 5 mL ddH_2_O containing 0.08% TFA. Loading of the samples (manual flow 1 mL/min) followed and columns were washed with 5 mL ddH_2_O. Elution of the compounds was performed with 4 mL methanol and the eluates were evaporated to dryness (rotary evaporator, 40 °C). Acetonitrile (1 mL) was added to the dried samples, and they were stored at −20 °C, protected from light. During the analyses all tubes were protected from light to avoid photodegradation of the analytes BPA and 4-tOP.

For the extraction of the analytes from the culture medium of *Artemia* nauplii, a 10 mL sample of the culture medium was used, its pH value was adjusted to 3.0 ± 0.1 using a 10% water solution of TFA (*v*/*v*) and the internal standard was added (75 μg/mL). Following the addition of 2 mL methanol, the sample was centrifuged for 10 min at 10,000 rpm and the supernatant was submitted to SPE, as described for biological samples.

### 2.5. Pollutant Administration

For the feeding assays, nauplii cultures (1000 mL cultures in filtered and autoclaved artificial seawater, salinity 35 ppt) [72,73] were fasted for 6 h and then Bisphenol-A or 4-tert-octylphenol was administered at 72 h, at a final concentration of each substance in each culture of 10 μg/mL and 0.5 μg/mL, respectively. The cultures were then incubated for 24 h under natural light. At the end of the incubation the nauplii and their culture medium were separated by filtration and stored at −20 °C.

### 2.6. Photodegradation Assay

Experiments on the potential photodegradation of the analytes were performed in glass tubes containing sterile seawater and BPA or t-OCT at a concentration of 50 μg/mL and 10 μg/mL, respectively. The tubes were incubated under intense sunlight for a total of 48 h. The concentration of the analytes BPA and t-OCT was determined at the onset (t = 0), at t = 24 h and at the end of the incubation period (t = 48 h), following extraction of the analytes.

### 2.7. Chromatography

The HPLC system comprised of an LC20_AD_ pump and an SPD-20A photodiode array detector (DAD) (Shimadzu, Kyoto, Japan), a Rheodyne 7125 injection valve, with a loop of 80 μL volume (Rheodyne, Cotati, CA, USA) and a Nucleosil 100 C_18_ column (250 × 4.6 mm, 5 μm), (Macherey-Nagel GmbH & Co., Duren, Germany) equipped with a 10 × 4.6 mm I.D., Nucleosil C_18_ precolumn. Separation of BPA, t-OCT, and n-OCT was performed using a binary mobile phase system of mobile Phase A (95:5, *v*/*v*, 0.1 % TFA in H_2_O: Acetonitrile) and mobile Phase B (5:95, *v*/*v*, 0.08 % TFA in H_2_O: Acetonitrile) at room temperature, as previously reported [62] but with adjustments to the gradient, starting at 100% mobile Phase A and proceeding to an increasing mobile Phase B concentration to 20% over 2 min, 70% over 5 min, 90% at 7 min, and until the end of the analysis. Flow rate was 0.5 mL/min, and the analytes were detected at 220 nm.

## 3. Results and Discussion

### 3.1. Analytes Extraction Efficiency

*Artemia* metanauplii are a solid sample that require homogenization to achieve cell lysis, and this was performed by homogenization of the sample in methanol-H_2_O containing 0.08% (*v*/*v*) TFA and centrifugation. The previously described protocol for the extraction of BPA from bacterial cultures [62] was modified regarding the methanol aqueous phase ratio from 5:1 to about 1:3, since this amount of methanol produced better extraction of analytes. Since alkylphenols are characterized by low solubility in water compared to Bisphenol-A (Table 1), methanol contributed to the increased solubility of t-octylphenol and BPA. The use of ethyl acetate or water, instead of methanol-TFA, was also employed in trial experiments but compound recovery was deemed inadequate as it resulted in the coextraction of *Artemia* lipids and matrix interferences. In the present study, 0.2 g wet weight tissue, corresponding to 0.01 ± 0.003 g dry weight, were processed since they resulted in satisfactory chromatograms with no interfering peaks in the blank samples (Appendix A). Although microwave-assisted solvent extraction [74], as well as ultrasonic bath extraction [75], were reported for the extraction of bisphenol and alkylphenols, homogenization is common in biological samples [75,76].

Following the first methanol-water-TFA extraction step, the samples were subjected to SPE, an acknowledged technique for the selective sample preparation prior to analysis of endocrine disruptors in liquid or pre-extracted solid samples [75]. SPE has been previously used for the extraction of bisphenol from complex matrices, such as rat tissues [50], human serum [77], or bacterial cultures [18]. The commonly used sorbent is C18 [18,50,77] although the use of polymeric sorbents [59] and silica gel [78] have been reported. However, C18 cartridges provided the cleanest extracts of fish samples compared to polymeric sorbents, employed in the analysis of bisphenol and alkylphenols [55]. A flow rate of 1 mL/min during the elution of SPE, and a water bath temperature lower than 40 °C during the evaporation of methanol, after SPE, were employed as previously recommended for bacteria samples [62] to facilitate the effective recovery of both analytes and the internal standard. However, it should be noted that although simple centrifuging and filtering was sufficient for medium samples originating from bacterial cultures in basal salt medium [62], this was not the case with the *Artemia* medium samples, since it resulted in clogging of the injection loop, due to the higher salt concentration of the sample. Hence, the SPE step was critical for efficient analysis.

The extraction in water-methanol and the following SPE step resulted in excellent recovery of BPA (100.4 ± 3.1) and of t-OCT (100.6 ± 3.9) in the case of the *Artemia* samples. The mean recovery in culture medium samples amounted to 99.9 ± 0.8 for BPA, while for t-OCT it amounted to 101.7 ± 1.9 (Table 2). The recovery values reported in the present work are higher to the previously reported values of 83.7 % (BPA) and 87.4 % (t-OP), for water extraction of Bisphenol-A and alkylphenols from zooplankton samples using Oasis HLB glass cartridges [27]. Our results for BPA and t-OCT extraction are in accordance with a recently published report on a water-acetonitrile extraction of bisphenols and octylphenols from fish and solid matrices, in which affinity chromatography clean-up instead of the reversed phase C_18_-SPE was employed and lower concentrations of 1 ng/g were used [79], resulting in recoveries of up to 101.1% for BPA and to 87.9 for t-octylphenol.

### 3.2. Method Validation

The results of the linear regression analysis performed are presented below (Table 2). The use of an internal standard was deemed necessary since sample matrix complexity, due to the presence of proteins and lipids in both samples, required an extraction protocol for both tissues and culture medium. 

To this end, 4-n-octylphenol was selected, since it has a similar structure to both bisphenol and tert-octylphenol, is well resolved from both analytes, and is not present in the blank samples (Appendix A).

Linearity of the standard curves in the *Artemia* culture medium and *Artemia* metanauplii (Table 2) was excellent, as indicated by the R^2^ values of 0.999 for BPA and t-OCT. The Relative Bias estimated as the percentage of ((calculated- nominal)/nominal]) was within the accepted range of ±10%. Intra-day precision was calculated by three analyses performed on the same day, while inter-day precision was calculated by performing six analyses over three days and estimated as % RSD, never exceeding 5.0% for all analytes. Our results indicate that the method is precise and repeatable with excellent recovery rates. The nominal concentrations of 1.0 μg/mL BPA and 1.0 μg/mL t-OCT in the culture medium were below the calculated LOD values (Table 3) and were, thus, excluded from the estimation of recovery results described above. The values for LOD and LOQ calculation were based on the response and the slope of the calibration curve.

The value of a LOD of 0.21 μg/mL and 0.17 μg/mL were estimated for BPA and t-OCT, respectively, in *Artemia* tissue corresponding to 10.5 ng/g dry weight and 8.6 ng/g dry weight, respectively, and of 0.63 μg/mL and 0.41 μg/mL for *Artemia* culture medium (Table 3). Although LOD values of 0.07 μg/kg were reported for BPA in mussels’ tissue [80], estimations were made with σ taken as the standard deviation of three samples spiked at the experimental estimated LOQ, and S as the slope of the corresponding calibration curve. On the other hand, LOD values of 5.6 μg/mL and 6.3 μg/mL, were reported following liquid–liquid extraction of water samples [81]. The previously reported LOQ values in zooplankton amounted to 2 ng/g dry weight for BPA and 0.8 ng/g dry weight for t-OCT, and to 5 and 1 ng/L, respectively, for water samples, determined as a tenfold signal-to-noise ratio for a sample with a very low analyte content [27], employing different methods for the biological and water samples. Variable results were obtained from the computation of LOD and LOQ values, depending on the statistical approach used. The residual standard deviation or error of intercept of calibration line was recommended instead of the mean blank signal value, as it is considered that it provides a more accurate estimate [82].

Since the LOD and LOQ values are calculated based on the calibration curve, they greatly depend on its range and lower values are expected when a range of lower concentrations are used. However, the range of the calibration curve should extend over the range of expected analytes’ concentrations in the samples [83]. In the present work, the expected concentrations of Bisphenol-A and t-octylphenol were within the range of the LC_50_ values. These values were acquired from the toxicity experiments that allowed the estimation of the amount that could be administered to *Artemia* metanauplii without leading to high mortality rates.

Typical HPLC chromatograms of *Artemia* and culture medium samples spiked with BPA, t-OCT, and n-OCT are presented in Figure 1. The peaks of the analytes BPA, t-OCT, and of the internal standard n-OCT, showed with a retention time (Rt) of 15.200 ± 0.12, 21.876 ± 0.11 26.241 ± 0.1 min, respectively. The peak appearing from 0 to 6 min, in all samples, possibly corresponds to the solvents, while the peak appearing at approximately 7.0 ± 0.5 min only in the medium samples possibly corresponds to medium constituents. Similar peaks also appear in the blank samples without, however, interfering with analysis (Appendix A).

### 3.3. Administration of Pollutants

BPA was administered at a concentration of 10 μg/mL, while t-OCT was administered at 0.5 μg/mL, since their LOD values amounted to15.6 and 0.73 μg/mL, respectively (Appendix A). The simultaneous administration of the two pollutants at LC_50_ levels was not applicable, since it resulted in higher mortality values, possible due to synergistic effects. Hence, each pollutant, namely BPA or t-OCT, was administered separately. Acute toxicity and inhibition of growth of the early developmental stages of *Artemia* nauplii (Instar I-II) have been reported after a 24 h exposure to BPA with an LC_50_ of 44.8 mg/L [43] for *Artemia franciscana*. The LC_50_ values for *Artemia sinica* nauplii amounted to 47.5 mg/L for BPA and to 2.27 mg/L for t-OCT [84]. On the other hand, the LC_50_ values of 70.1 μg/L and 7.7 μg/L were reported for adult (15 days old) *Artemia sinica* individuals exposed to BPA and n-hexylphenol [45], illustrating that the advanced developmental stages present greater sensitivity to these pollutants. The increased sensitivity of subsequent developmental stages to pollutants was attributed to the fact that the organisms appear more ravenous compared to the earlier developmental stages, during which the animals depend on reserved yolk for their energy requirements [85]. In addition to the importance of the developmental stage employed, the origin of cysts and abiotic parameters regarding, among others, the pH, hatching temperature, photoperiod duration, and saltwater treatments are also of great importance [42]. Considering that bacteria are capable of BPA biotransformation [18], the use of axenic cultures in the autoclaved medium were used in the present study. Representative chromatograms of the pollutants following administration of 10 μg/mL BPA or 0.5 μg/mL t-OCT to *Artemia*, that are below their estimated LC_50_ values, are presented in Figure 2 for *Artemia*, and in Figure 3 for its culture medium.

The peaks appearing at 12.5 ± 0.18, 13.1 ± 0.08 and 13.8 ± 0.2 and apparent following administration of either pollutant (Figure 2A,B) might correspond to products produced during their biotransformation by *Artemia*. Similar peaks were observed following BPA microbial biotransformation corresponding to 4-Hydroxyacetophenone, 4-Hydroxybenzoic acid, and hydroquinone [18]. Moreover, biodegradation of tert-octylphenol by the fungus *Thielavia* sp. HJ22 has been reported to result in the formation of 4-hydroxybenzoic acid hydroquinone [86]. Although UV spectra presented a good match with standard compounds, the identity of these peaks was not confirmed by MS in the present work. The peak that appears at 19.7 (Figure 2B and Figure 3B) is evident only following administration of t-OCT and might correspond to a photodegradation product, since similar peaks were observed during the photodegradation experiment (Appendix A). Photodegradation via photolysis has been confirmed for BPA [25,27] and for t-OCT [87], and the products of solar transformation presented higher toxicity than the parent compound [88]. The feeding experiments were conducted under normal light and, hence, photodegradation losses were estimated to examine whether any pollutant concentration decline is due to photolysis. In the photolysis experiments, the initial BPA and t-OCT concentration were calculated at 49.15 μg/mL and 9.81 μg/mL, respectively, and their concentration following a 24 h incubation amounted to 44.63 μg/mL and 7.87 μg/mL, while after 48 h BPA amounted to 37.94 and t-OCT to only 1.7 μg/mL. Hence, the reduction in the analyte concentration induced by photodegradation was 9.2% for BPA and 19.7% for t-OCT at 24 h (Table 4).

These values are in accordance with those reported for bisphenol A degradation [89] but higher than the previously reported absence of t-OCT degradation under one hour of direct solar irradiation [90]. Although part of the pollutants is degraded via photolysis, this does not compensate for the observed reduction in the total levels of both pollutants (Table 4). The amounts detected in the processed medium samples were 66.8 and 3.0 μg and the total amounts in a 1000 mL culture corresponded for BPA to 66.8% and for t-OCT to 63.8% of the initially added amounts. In addition, the amounts detected in *Artemia* tissue amounted to 4.1% and 0.7% of the initially added amounts for BPA and t-OCT, respectively, and a reduction of up to approximately 32% in total BPA and of 35% in total t-OCT was observed in our experiments. The accumulation of BPA in zooplankton has been confirmed in an eco-system bioreactor during incubation of phytoplankton (*Nannochloropsis* sp.) and zooplankton (*Artemia* sp. or *Brachinous* sp.) in the presence of the pollutant [91].

The occurrence of high levels of BPA and alkylphenols in the marine environment has been documented in aquaculture facilities which are in the vicinity of industrial or urban activity [92]. The BPA values amounted to 37 ng/L in mussels cultured in Thailand [92], to 1.5 ng/g d. w. in fish cultured in Taiwan [93], to 4.2 ng/g in the muscle of fish cultured in Malaysia [94], or 272 ng/g d. w. in the North Atlantic Ocean [24]. The relevant t- octylphenol concentration was 16 ng/ g d. w. in mussels cultured in Malaysia [94], while the occurrence of the alkylphenol nonylphenol has been documented in the sediments of the Northern Aegean, indicating a risk for the organisms that live in it [95]. The cultured fish fry fed on *Artemia* have a weight of 15 mg and consume about 300 nauplii per day [96], corresponding to a wet weight of 5 mg, indicating that these pollutants, if present in *Artemia*, may accumulate and possibly pose a health risk to the fish larvae. Recently the accumulation of bisphenols in farmed seabream muscle following exposure to microplastics, amounting to 0.3 μg/g muscle, was reported and the authors consider aquaculture equipment as a source of such microplastics [97]. Considering the TDI for BPA amounting to 0.04 ng/kg b. w. daily and the suggested TDI for t-OCT for men of 0.067 ng/kg b. w. daily (Table 1), it becomes apparent that the exposure of cultured organisms to the pollutants might also pose a possible hazard to human health. The use of marine phytoplanktons and zooplanktons has been suggested for the recovery of BPA from seawater [91]. The uncompensated decline in both BPA and t-OCT levels observed in the present study indicates that, although these compounds might accumulate in *Artemia*, their biotransformation is possible. Since the toxicity of the possible biotransformation products has not been investigated and *Artemia* is used as a live feed for cultured marine larvae, the monitoring of its pollutants levels is of extreme importance.

## 4. Conclusions

The developed analytical method provides a simple approach for the simultaneous detection and quantification of bisphenol A and t-octylphenol in the live fish feed *Artemia* and in its culture medium. The presented extraction protocol and the use of 4-n-octylphenol as the internal standard provided a detection limit of 0.21 for BPA and 0.17 ng/μL for t-OCT. The method was applied in the analysis following administration of the pollutants to *Artemia* metanauplii at levels close to their corresponding LC_50_. The administration experiments revealed that *Artemia* is capable to uptake bisphenol and t-octylphenol from the culture medium. Experimental efforts to reduce the pollutant load of *Artemia* are currently under investigation. Further experiments are needed to evaluate the transfer of the pollutants to the fish larvae that feed on *Artemia*.

## Figures and Tables

**Figure 1 mps-05-00038-f001:**
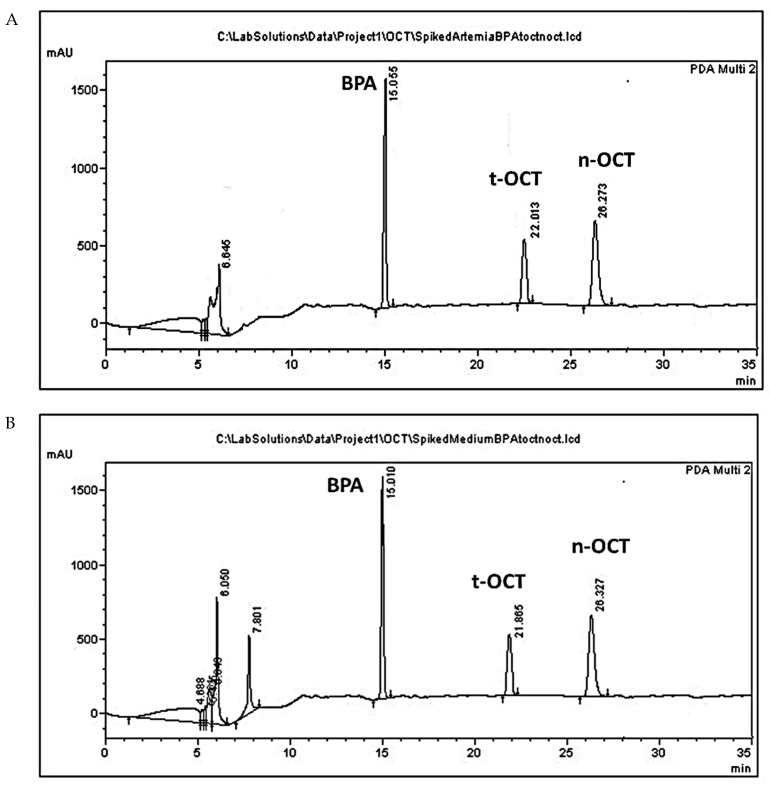
HPLC chromatograms of (**A**) *Artemia* tissue; and (**B**) culture medium samples containing BPA (50 μg/mL), t-OCT (10 μg/mL) and the internal standard n-OCT (75 μg/mL).

**Figure 2 mps-05-00038-f002:**
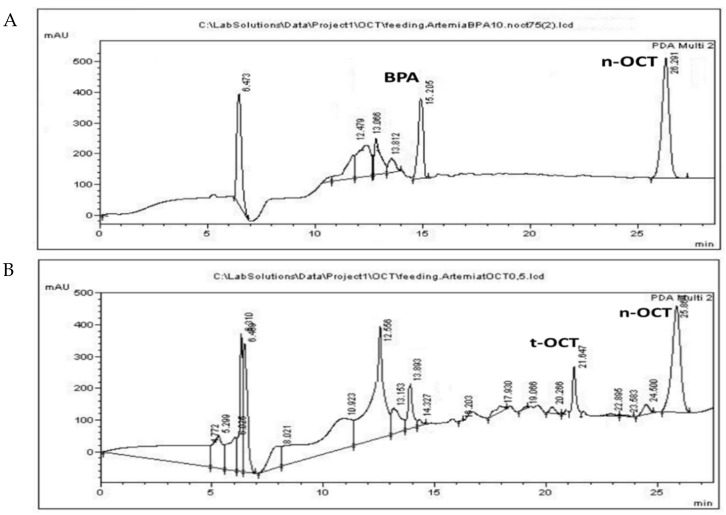
HPLC chromatograms of *Artemia* tissue following administration of (**A**) 10 μg/mL BPA; and (**B**) 0.5 μg/mL t-OCT.

**Figure 3 mps-05-00038-f003:**
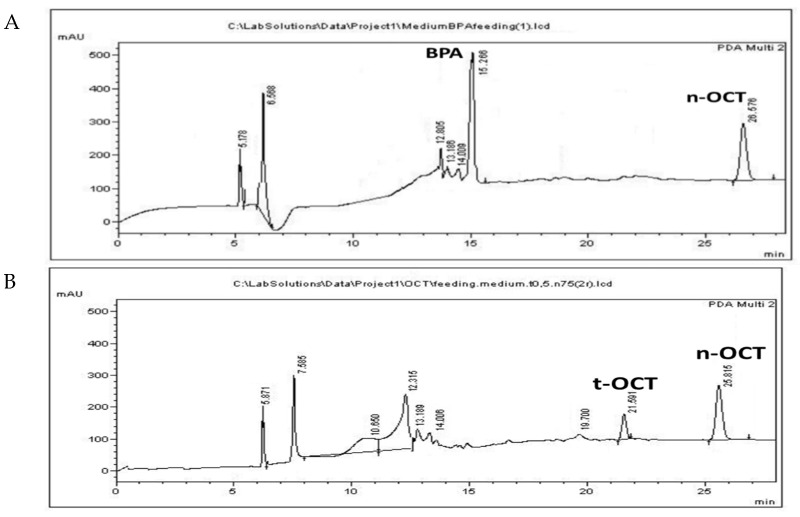
HPLC chromatograms of culture medium following the administration of (**A**) 10 μg/mL BPA; and (**B**) 0.5 μg/mL t-OCT.

**Table 1 mps-05-00038-t001:** Compounds investigated and characteristic parameters.

Compound (Abbreviation, CAS#, Formula)	Structure	MW	Solubility in Water (mg L^−^^1^)	LD50 g kg^−1^	TDI (Tolerable Daily Intake) ng/kg bw/day
Bisphenol-A BPA, 80-05-7 C_15_H_16_O_2_	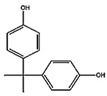	240.20	120 [66]	3-5 (rat) [67]	4000 (human) [68]
4-tert-Octylphenol t-OCT, 140-66-9 C_14_H_22_O	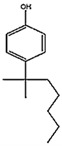	206.32	5.1 [69]	4.6 (rat) [69]	0.067 (men) 33.3 (women) [70]
4-n-Octylphenol n-OCT, 1806-26-4 C_14_H_22_O	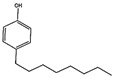	206.32	3.1 [71]	87.8 µg L^−1^ (fish) [71]	-

**Table 2 mps-05-00038-t002:** Linearity, accuracy (bias), precision, and recovery of BPA and t-OCT determination in the tissue and the culture medium of *Artemia*.

Analyte Sample Std. Curve /R^2^	Nominal Conc. (μg/mL)	Calculated Conc. (μg/mL) (Mean ± SD)	Relative Bias (%)	Precision		Recovery (%)	
				Intra- (*n* = 3)	Inter- (*n* = 2 × 3)	(Mean ± SD)	RSD%
BPA culture medium y = 0.0027x − 0.024 0.9998	1	0.94 ± 0.002	−5.3	0.2	0.3	94.8 ± 0.3	0.3
	25	24.86 ± 0.1	−0.6	0.4	0.5	99.4 ± 0.4	0.4
	50	50.6 ± 0.3	1.2	0.6	0.7	101.2 ± 0.6	0.6
	75	74.8 ± 0.09	−0.2	0.1	0.2	99.8 ± 0.1	0.1
	100	100.2 ± 0.6	0.2	0.6	0.7	100.2 ± 0.6	0.6
	125	123.7 ± 1.8	−1.1	0.4	1.4	98.9 ± 1.4	1.4
BPA *Artemia* y = 0.045x − 0.0887 0.999	2.5	2.64 ± 0.02	5.9	0.4	0.6	105.6 ± 0.6	0.6
	5	4.81 ± 0.05	−3.8	0.8	1.0	96.3 ± 0.9	0.9
	7.5	7.61 ± 0.02	1.3	0.1	0.3	101.4 ± 0.3	0.3
	10	9.86 ± 0.02	0.1	0.2	0.7	98.6 ± 0.2	0.2
	12.5	12.45 ± 0.03	−0.4	0.1	0.2	99.6 ± 0.2	0.2
	15	15.11 ± 0.02	0.7	0.03	0.1	100.7 ± 0.1	0.1
t-OCT culture medium y = 0.018x + 0.037 0.999	1	1.03 ± 0.04	3.1	0.6	4.3	103.1 ± 4.5	4.3
	2.5	2.60 ± 0.04	4.1	0.5	1.6	104.1 ± 1.7	1.6
	5	5.05 ± 0.03	1.1	0.03	0.7	101.1 ± 0.7	0.7
	7.5	7.69 ± 0.02	2.5	0.1	0.2	102.5 ± 0.2	0.2
	10	9.87 ± 0.03	−1.2	0.1	0.3	98.8 ± 0.3	0.3
	12.5	12.75 ± 0.1	2.0	0.7	0.9	102.0± 0.9	0.9
t-OCT *Artemia* y = 0.019x + 0.012 0.9989	1	1.08 ± 0.02	8.7	0.01	1.9	108.7 ± 2.0	1.9
	2.5	2.41 ± 0.01	−3.4	0.5	0.6	96.6 ± 0.6	0.6
	5	4.85 ± 0.09	−2.8	0.4	1.8	97.2 ± 1.8	1.8
	7.5	7.59 ± 0.03	1.3	0.1	0.4	101.3 ± 0.4	0.4
	10	9.98 ± 0.02	−0.2	0.3	0.3	99.8 ± 0.2	0.2
	12.5	12.49± 0.2	0.2	0.4	1.8	100.2 ± 1.8	1.8

**Table 3 mps-05-00038-t003:** LOD, LOQ, and RT values for BPA and t-OCT.

Std. Curve	Retention Time (min) Mean ± SD (RSD %)	LOD (μg/mL)	LOQ (μg/mL)
BPA in *Artemia*	15.200 ± 0.12 (0.82)	0.21	0.65
BPA in culture medium	15.310 ± 0.10 (0.69)	0.63	1.92
t-OCT in *Artemia*	21.876 ± 0.11 (0.49)	0.17	0.52
t-OCT in culture medium	21.597 ± 0.31 (1.46)	0.41	1.25

**Table 4 mps-05-00038-t004:** Bisphenol A and t-octylphenol levels in administration experiments.

Compound	BPA	t-OCT
Administered in 1000 mL culture (μg)	10,000.0	500.0
Found in *Artemia* (μg/g wet weight)	68.3	5.9
Found in culture medium (μg/mL)	6.68	0.3
Total found in *Artemia* (μg)	40.9	3.6
Total found in 1000 mL medium (μg)	6680	319.3
Total found (μg)	6720.9	322.9
Concentration reduction in cultures (%)	32.8	35.4
Decline due to Photolysis (%)	9.2	19.7

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
