# Peer review of "Simultaneous Quantification of Bisphenol-A and 4-Tert-Octylphenol in the Live Aquaculture Feed Artemia franciscana and in Its Culture Medium Using HPLC-DAD"

_mps, 2022, doi:10.3390/mps5030038_

Round 1
Reviewer 1 Report
In this work, Giamaki et al. states the development and validation of an analytical method to determine BPA and tert-octylphenol in Artemia and in Artemia culture medium.
The title is representative and the abstract concise and clear enough. The vague statement from the Abstract has to be clarified "a reduction in their levels in Artemia culture media up to 35% was observed and could not be accounted for by photodegradation.."
Introduction
Line 36-47: State the percentage of BPA and t-OCT in plastics.
State the current BPA legally admitted dosage and the controversy that's currently taking place "BPA exposure and toxicity in January 2015 and reduced the tolerable daily intake (TDI) from 50 to 4 µg/kg bw."; "In 2018, PlasticsEurope filed a legal action against the decision of the European Chemicals Agency (ECHA) to identify bisphenol A (BPA) as a substance of very high concern" [1,2].
Data regarding how BPA is metabolised [3] should be stated.
Data regarding bacterial growth enhancing effect and oxidative stress induction of BPA in organisms in aqueous environment has to be stated [4].
Although banned in some countries such as Canada, in Europe the fight for BPA to stay on shelves remains, therefore such an aspects has to be stated in BPA related scientific publications.
Materials and methods
Line 122: Clarify " bisphenol.... or 4-tert-octylphenol.." clarify this statement.
Line 236: missing reference error message.
References
- https://www.efsa.europa.eu/en/topics/topic/bisphenol
- https://plasticseurope.org/wp-content/uploads/2021/10/20210301-BPA-Judgment-appeal.pdf
- Nachman, R. M., Hartle, J. C., Lees, P. S., & Groopman, J. D. (2014). Early Life Metabolism of Bisphenol A: A Systematic Review of the Literature. Current environmental health reports, 1(1), 90–100. https://doi.org/10.1007/s40572-013-0003-7
- Pop, C.-E.; Draga, S.; Măciucă, R.; Niță, R.; Crăciun, N.; Wolff, R. Bisphenol A Effects in Aqueous Environment on Lemna minor. Processes 2021, 9, 1512. https://doi.org/10.3390/pr9091512
Author Response
Thank you for your prompt and thorough response. You will find in the attached file that we confronted all issues raised and amended the relevant parts in accordance with your suggestions

Reviewer 2 Report
The authors describe the analytical HPLC method for the simultaneous detection and quantification of bisphenol A and t-octylphenol in live fish feed Artemia and in its culture medium. They validated the method, set the LOD and LOQ values and describe the extraction procedure of analytes from solid samples.
I have the following questions and comments:
Major comments
Introduction: the pragraph about the analytical methods (l.74-83)-the authors cite the analytical methods which deals with the determination of BPA and alkylphenols but it is not clear if they are hplc methods or not. It is time-consuming for readers to find all the citations. I propose to indicate what method it is (I suppose that for the purposes of this article, the authors limited themselves to hplc determination), the type of stationary phase, eluent and the detection, the merits and shortcomings of these methods and why they cannot be used for the work presented. So that the reader has an overview of what types of columns, mobile phases and detection methods are used to determine the relevant substances. This section is underestimated compared to the rest of the text in the introduction.
Maybe I overlooked it, but I wonder what is the real concentration of the determined substances in real aquaculture. Please mention in the discussion.
Other minor comments
Please use consistently established abbreviations throughout the text (t-OCT in l.47 or 50 is only OCT.
Please use consistently italics for the genus Artemia and not for Artemia culture, tissue etc. (for example in l. 245) thorough the text and tables.
Table 1-Formulas are somehow rotated; TDI-for whom
Table 2-try to change the name of the table to better describe the content
l.236-probably the reference to Table 2 should be
In all pictures where there are chromatograms, the title should be "HPLC chromatogram of ....."
Table 4-the numbers are not in columns; total found-units are missing
Author Response

(The authors gave the same response as above.)

Round 2
Reviewer 1 Report
The authors have addressed all my remarks.